# Adipokines and Gamma-Glutamyl Transferase as Biomarkers of Metabolic Syndrome Risk in Mexican School-Aged Children

**DOI:** 10.3390/nu16244410

**Published:** 2024-12-23

**Authors:** Elizabeth Solís-Pérez, Ana Marina Mar-Buruato, Alexandra Tijerina-Sáenz, Maria Alejandra Sánchez-Peña, Blanca Edelia González-Martínez, Fernando Javier Lavalle-González, Jesús Zacarías Villarreal-Pérez, Gerardo Sánchez-Solís, Manuel López-Cabanillas Lomelí

**Affiliations:** 1Facultad de Salud Pública y Nutrición (FaSPyN), Universidad Autónoma de Nuevo León (UANL), Monterrey 64460, Nuevo León, Mexico; elizabeth.solis@uanl.mx (E.S.-P.); ana.marbr@uanl.edu.mx (A.M.M.-B.); alexandra.tijerinas@uanl.mx (A.T.-S.); maria.sanchezpn@uanl.edu.mx (M.A.S.-P.); blanca.gonzalezma@uanl.mx (B.E.G.-M.); 2Universidad Autónoma de Nuevo León (UANL), Hospital Universitario “Dr. José Eleuterio González”, Servicio de Endocrinología, Monterrey 64460, Nuevo León, Mexico; fernando.lavallegn@uanl.edu.mx (F.J.L.-G.); zacvilla@yahoo.com.mx (J.Z.V.-P.); gerardo.sanchezsls@uanl.edu.mx (G.S.-S.)

**Keywords:** adipokines, gamma-glutamyl transferase (GGT), metabolic syndrome, children

## Abstract

**Background/Objectives**: The prevalence of metabolic syndrome in children has been increasing, raising concerns about early detection and clinical management. Adipokines, which are secreted by adipose tissue, play a critical role in metabolic regulation and inflammation, while gamma-glutamyl transferase (GGT), as a liver enzyme, is linked to oxidative stress and metabolic disorders. The objective was to examine the association of circulating adipokines and GGT with metabolic syndrome risk in school-aged children from Northeast Mexico. **Methods**: A total of 140 children from 6 to 12 years of age in the state of Nuevo León, Mexico, participated in this study. Obesity was classified according to the BMI z-score by the World Health Organization (WHO, 2007), and metabolic syndrome was classified according to the International Diabetes Federation (IDF, 2007). Serum levels of leptin, adiponectin, TNF-α, IL-6, and GGT were measured. Statistical analysis was performed using the Student’s *t*-test, simple linear regression analysis, and receiver operating characteristic (ROC) curve analysis. **Results**: Among the 140 participants, 60 children (43%) were classified as obese, and of those children with obesity, 55% were diagnosed with metabolic syndrome. Leptin was significantly associated with waist circumference (WC), systolic blood pressure (SBP), serum glucose, triglycerides, and HDL cholesterol (HDL-c). Adiponectin also showed significant associations with WC, SBP, serum triglycerides, and HDL-c. GGT was significantly correlated with WC and HDL-c, while IL-6 and TNF-α did not indicate significance. Associations were observed among leptin, adiponectin, and GGT, highlighting their combined role as potential markers for metabolic syndrome in children. The ROC curve analysis and Youden’s index provided cut-off points for these biomarkers: leptin: 8.3665 ng/mL, adiponectin: 9.87 µg/mL, GGT: 17.8 U/L, IL-6 2.77 pg/mL, and TNF-α: 6.68 pg/mL; **Conclusions**: These findings emphasize the utility of leptin, adiponectin, and GGT as early biomarkers for identifying children with obesity who are at risk of developing metabolic syndrome.

## 1. Introduction

In the past years, diseases such as obesity and metabolic syndrome have significantly increased in children and adolescents in Mexico. The prevalence of combined overweight and obesity in children 5 to 11 years of age was 34.4% in 2012, 35.5% from 2018 to 2019 [1], and 36.5% from 2020 to 2024, while the prevalence of obesity itself was 17.5% [2]. In the state of Nuevo León, data from ENSANUT Continua 2022 revealed a prevalence of combined overweight and obesity in children 5 to 11 years old of 34.2%, in which girls had a higher incidence, at 36.8%, than boys, at 31.8%. The prevalence of obesity was reported as 14.8% in boys and 24.9% in girls [3]. Metabolic syndrome represents a significant health problem, with a world incidence between 15% and 40% of the general population, raising concerns about early detection and clinical management [4,5]. In Mexico, there is no precise information regarding the prevalence in children of school age, although a recent study showed that 58.81% of children and adolescents presented with metabolic syndrome according to the de Ferranti criteria [6]. In addition, the previous literature demonstrated that around 40 to 48% of boys and 50 to 60% of girls met the WHO criteria for metabolic syndrome, while 33% of boys and 42% of girls met the IDF criteria [7]. The presence of metabolic syndrome is mainly due to general lifestyle factors, such as poor food consumption habits and the lack of physical activity, mainly during childhood ages [8,9].

Obesity is related to insulin resistance and metabolic syndrome, both of which are associated with elevated levels of gamma-glutamyl transferase (GGT) and altered levels of adipokines. GGT is an enzymatic marker of hepatic dysfunction that is related to the risk of developing metabolic syndrome and thus may be used as a predictive marker for the disease. Several longitudinal studies have reported a relationship between an elevated serum GGT and the incidence of metabolic syndrome and associated factors, such as obesity, pre-hypertension, insulin resistance, and type-2 diabetes (T2D); elevated serum GGT may also be a risk factor for the development of metabolic dysfunction-associated steatotic liver diseases [10,11,12]. High levels of GGT may reflect high oxidative stress levels, considering it as a marker of metabolic syndrome, thus explaining the association between diseases due to a chronic inflammatory condition of low degree and oxidative stress [13,14]. Therefore, serum GGT may be used as an inflammatory marker and oxidative stress and could be useful for the diagnosis of metabolic syndrome in the childhood population [10].

Adipose tissue-secreted adipokines are a group of bioactive molecules that play critical roles in regulating different physiological and pathological processes, such as inflammation, metabolism, appetite, cardiovascular function, immunity, and insulin sensitivity, among others [15,16]. Insulin resistance is a condition whereby insulin-induced glucose uptake is impaired in the insulin sensitivity tissue [17]. Adipokines are a key factor between obesity and the components of metabolic syndrome and play an important role in their development. These include leptin and adiponectin, tumoral necrosis factor alpha (TNF-α), and interleukin 6 (IL-6). Leptin and adiponectin are directly related to body fat levels, and they have an effect on lipid and glucose metabolism [18]. Adiponectin regulates insulin sensitivity, and it has anti-inflammatory, antioxidant, and cardioprotective properties; thus, low levels of this adipokine have an important role in the development of metabolic syndrome. On the contrary, elevated levels of leptin contribute to insulin resistance and dysfunction in glucose metabolism. Interleukin 6 (IL-6) levels increase in individuals with obesity as there is greater expression in increased visceral fat tissue. It is considered a pro-inflammatory marker related to insulin resistance and glucose intolerance. TNF-α increases insulin resistance and promotes the release of fatty acids from the adipose tissue to the systemic circulation, thereby playing an important role in the pathogenesis of metabolic syndrome [19,20].

Although evidence on the relationship between adipokines and metabolic syndrome in children in Mexico is growing, the age groups studied are diverse, and the association with GGT remains to be elucidated [6]. This study aims to examine the association of circulating adipokines and GGT with metabolic syndrome risk in school-aged children (6 to 12 years) in Northeast Mexico.

## 2. Materials and Methods

### 2.1. Study Population and Subject Selection

In this cross-sectional study, obese children were recruited from the University Nutrition Program of Childhood Obesity. For comparison, children without obesity were also recruited from the same community (schools in the metropolitan area of Monterrey, Nuevo León, Mexico). This study was approved by the Research and Ethics Committees of the UANL-Nutrition and Public Health School (Facultad de Salud Pública y Nutrición, Universidad Autónoma de Nuevo León) under protocol number 21-FASPYN-SA-23 TP, in accordance with the Health General Law of Mexico and the Declaration of Helsinki. Parents and children were informed about the characteristics and outcomes of this study, and they signed the consent and assent forms, respectively.

A non-probabilistic sample of 140 school-aged children 6 to 12 years old was considered. The study sample selection criteria are as follows: children from the state of Nuevo León, minimum age of 6 years, and maximum age of 12 years and 11 months.

The BMI z-score (WHO, 2007) was used to group children into normal weightand obesity groups, as follows [normal −1 to +1 SD, obesity > +2 SD] [21], and no medical conditions were permitted in participants besides those of this study. The exclusion criteria are as follows: children from other regions outside of the state of Nuevo León because they do not attend the Program of Childhood Obesity, ages below 6 years and above 13 years, a BMI z-score of low weight [<−2 SD], and the presence of medical conditions other than those of this study. Children with incomplete data were eliminated from the study.

### 2.2. Anthropometric and Medical Assessment

A medical assessment was performed by a pediatrician and medical doctor interns in order to confirm the selection criteria. Anthropometric measurements were performed by registered nutritionists, including weight (kg), which was recorded to the nearest 100 g using a Tanita scale BC-533^®^; height (m), which was was measured with a stadiometer SECA 217^®^ to the nearest 1 cm while subjects were in a barefoot standing position with their shoulders in a normal position; and waist circumference (WC, cm), which was measured with a flexible steel tape SECA 203^®^ in a central point between the lowest rib and the iliac crest. Body mass index (BMI, kg/m^2^) was calculated as weight (kg) divided by squared height (m^2^). The BMI z-score used to group children into normal weight, overweight, and obesity were normal weight −1 to +1 SD, overweight > +1 and <+2 SD, and obesity > +2 SD [18]. Blood pressure (BP) was measured twice in the right arm of subjects who had been resting for at least 10 min in a seated position using a mercury sphygmomanometer. Taking into account the medical and nutritional assessments and records, a diagnosis for metabolic syndrome was defined based on the criteria of the International Diabetes Federation (IDF) (2007) [7].

### 2.3. Biochemical Measurements

Blood samples were obtained by clinical laboratory personnel specialized in pediatric patients. Blood extraction from the antecubital vein was obtained after a 12 h fast and centrifuged within 2 h for the separation of serum. The serum samples were sent for biochemical analyses to the General and Endocrinology Laboratories of the Hospital Universitario Dr. José Eleuterio González. Assays were performed in triplicate. The laboratories routinely monitor both inter- and intra-assay coefficients of variation for all assays with a goal of keeping themat 5% or below.

Enzymatic determinations were performed in an UniCel Dxc 600 analyzer (Beckman Coulter, Brea, CA, USA) using available commercial kits for triglycerides (mg/dL) via glycerol phosphate oxidase, HDL cholesterol (mg/dL) via hexokinase, glucose (mg/dL) via hexokinase, and gamma-glutamyl transferase (GGT) (U/L) via kinetic reaction. Adipokines: Leptin (ng/mL), adiponectin (µg/mL), tumor necrosis factor alpha (TNF-α) (pg/mL), and interleukin 6 (IL-6) (pg/mL) were analyzed via flow cytofluorometry using an RIA Assay Kit Luminex^®^ xMAP^®^.

### 2.4. Statistical Analysis

A normality test was run using the Kolmogorov–Smirnov test, obtaining normal distributions. Descriptive data were shown as median and standard deviation (SD), and categorical data were presented as frequency and percentage (n (%)). A Student’s *t*-test for independent samples was run to compare the clinical, biochemical, and anthropometric parameters between two groups of children (with vs. without obesity and those with vs. without metabolic syndrome) [22], and Levene’s test was considered for equal variance assumption. A chi-squared test was used to compare categorical data between groups.

A linear regression model was used to determine the association between adipokines and GGT and the presence of metabolic syndrome, thus possibly identifying the former as a predictive factor of the disease [23]. Lastly, a receiver operating characteristic (ROC) curve analysis was conducted; the area under the ROC curve (AUC) and Youden’s index were also estimated to establish reference values (cut-off points) of the markers in the study that may help predict the development of metabolic syndrome in children between 6 and 12 years of age [24].

Statistical analyses were conducted using SPSS^®^ Statistics version 25 (IBM^®^ SPSS^®^), considering a *p* value of <0.05 as significant.

## 3. Results

### 3.1. Nutritional Assessment

The study population was 140 children between 6 and 12 years of age; 57% of the total presented a normal weight, 43% had obesity, 83 were female (59%), and 57 were male, with a median age of 9 ± 1.87 years. Of those children with obesity (*n* = 60, 43%), 55% presented with metabolic syndrome (or 23.6% of the study population, 33 children out of 140). The clinical, anthropometric, and biochemical characteristics are presented in Table 1 in order to compare the differences between children with obesity and those of normal weight.

There were significant differences between these groups in height in cm (144.78 ± 11.57 vs. 131.98 ± 13.63; *p* < 0.001), body weight in kg (60.07 ± 14.37 vs. 29.87 ± 7.96; *p* < 0.001), BMI in kg/m^2^ (28.28 ± 3.89 vs. 16.78 ± 1.31; *p* < 0.001), BMI z-score (3.29 ± 0.86 vs. 0.22 ± 0.47; *p* < 0.001), waist circumference in cm (89.90 ± 11.09 vs. 58.42 ± 5.60; *p* < 0.001), SBP in mmHg (99.05 ± 10.51 vs. 88.99 ± 11.40; *p* < 0.001), triglycerides in mg/dL 175.92 ± 79.62 vs. 108.93 ± 34.25; *p* < 0.001), and HDL cholesterol in mg/dL (36.80 ± 7.91 vs. 52.01 ± 12.06; *p* < 0.001). There was no difference in DBP or glucose levels between groups.

Table 2 presents the characteristics of obese children with vs. without metabolic syndrome. The main significance is the difference between triglyceride levels in mg/dL (216.21 ± 80.16 vs. 126.67 ± 43.54; *p* < 0.001) and HDL cholesterol in mg/dL (32.67 ± 3.93 vs. 41.85 ± 8.65; *p* < 0.001). There were no differences between the groups in height, weight, z-score of BMI, WC, SBP, DBP, or glucose levels.

### 3.2. Adipokines and GGT Levels

When comparing the two groups, those with vs. those without obesity (Table 3), a significant difference in leptin levels in ng/mL (30.61 ± 18.60 vs. 4.14 ± 3.19; *p* < 0.001), adiponectin in µg/mL (29.54 ± 23.95 vs. 67.71 ± 41.63; *p* < 0.001), and GGT in U/L (21.86 ± 13.49 vs. 15.00 ± 2.69; *p* < 0.001) is observed. There were no statistical differences between IL-6 in pg/mL (5.11 ± 4.30 vs. 4.26 ± 5.30; *p* = 0.310) or TNF-α in pg/mL (6.64 ± 3.66 vs. 5.94 ± 4.47; *p* = 0.320). Overall, the levels of leptin, adiponectin, and GGT were significantly higher in children with obesity.

There was a significant difference between GGT levels in U/L (18.34 ± 7.77 vs. 26.17 ± 17.43; *p* = 0.020) in obese children with vs. without metabolic syndrome; on the other hand, there were no significant differences in leptin levels in ng/mL (32.42 ± 21.39 vs. 28.40 ± 14.58; *p* = 0.400), adiponectin in µg/mL (28.00 ± 19.98 vs. 31.42 ± 28.34; *p* = 0.580), IL-6 in pg/mL (5.53 ± 4.90 vs. 4.59 ± 3.45; *p* = 0.400), or TNF-α in pg/mL (6.71 ± 4.15 vs. 6.55 ± 3.04; *p* = 0.870) between the groups (Table 4).

### 3.3. Association of Markers with Metabolic Syndrome

A linear regression analysis was used to evaluate the association of adipokines and GGT with metabolic syndrome, demonstrating that 27% of the variance was explained by leptin, while adiponectin explained 10% of the model variance. Adipokines, IL-6, TNF-α, and GGT exhibited no statistical effect on model variance (Table 5).

We also analyzed adipokines and GGT with the components of metabolic syndrome. Leptin was associated with WC (*p* < 0.001), SBP (*p* < 0.001), glucose levels (*p* = 0.026), triglycerides (*p* < 0.001), and HDL cholesterol *(p* < 0.001). Adiponectin showed an association with WC (*p* < 0.001), SBP (*p* < 0.001), triglycerides (*p* < 0.001), and HDL cholesterol (*p* < 0.001); GGT was associated with WC (*p* < 0.001) and HDL cholesterol (*p* = 0.006) (Table 6). 

IL-6 was significantly associated with HDL cholesterol (*p* = 0.036). TNF-α showed no significant association. DBP was not associated with adipokines or GGT (Table 7).

### 3.4. Association Among Adipokines and GGT Levels

Levels of adipokines and GGT were analyzed to determine specific associations using a linear regression model (Table 8). Leptin presented a significant association with adiponectin (R^2^ = 0.120, *p* < 0.001) and GGT (R^2^ = 0.113, *p* < 0.001). Adiponectin was associated with GGT (R^2^ = 0.088, *p* < 0.001), and IL-6 and TNF-α were also associated (R^2^ = 0.092, *p* < 0.001).

### 3.5. Cut-Off Points for Circulating Adipokines and GGT to Predict the Development of Metabolic Syndrome in Children

A receiver operating characteristic (ROC) curve analysis and area under the ROC curve (AUC) were conducted for each adipokine and GGT (Figure 1, Table 9) to determine cut-off points for the prediction of metabolic syndrome in children. Areas under the ROC curve (AUC) were 0.833 for leptin (CI 95%, 0.747–0.918, *p* < 0.001), 0.243 for adiponectin (CI 95%, 0.157–0.328, *p* < 0.001), 0.626 for GGT (CI 95%, 0.508–0.744, *p* = 0.029), 0.647 for IL-6 (CI 95%, 0.552–0.743, *p* = 0.011), and 0.552 for TNF-α (CI 95%, 0.437–0.667, *p* = 0.059).

Table 10 indicates the proposed cut-off points of serum markers as risk levels for presenting metabolic syndrome in children (leptin: 8.3665 ng/mL, adiponectin: 9.87 µg/mL, GGT: 17.8 U/L, IL-6: 2.77 pg/mL, and TNF-α: 6.68 pg/mL), according to Youden’s index.

## 4. Discussion

Alarming data on obesity and metabolic syndrome and a growing incidence in very young individuals, such as school-aged children, have elevated these issues to health problems of high relevance for professionals and researchers in multidisciplinary settings. The identification of nontraditional biomarkers in clinical practice is pertinent for the management and prevention of complications in children with these conditions. Therefore, this concern has motivated us to examine the association between circulating leptin, adiponectin, and GGT and metabolic syndrome in school-aged children from Northeast Mexico.

When comparing the clinical, anthropometric, and biochemical characteristics of children with obesity with those of children with normal BMIs (with normal weight), all the parameters were significantly different, except for glucose levels (84.83 vs. 82.79 mg/mL (*p* = 0.09)) and diastolic blood pressure (64.20 vs. 62.06 mmHg (*p* = 0.10). This discrepancy may reflect the younger age group (6–12 years), in which insulin resistance and hyperglycemia might not yet be prominent due to shorter obesity duration. Most findings confirm the established evidence; however, the lack of glucose elevation may emphasize the importance of early intervention before metabolic dysfunction becomes more pronounced in older children or adolescents [25,26].

In our population, height was greater in children in the group with obesity (144.78 cm) vs. that of normal-weight children (131.98 cm), and children with obesity presented twice the body weight of those with normal weight (60.07 kg vs. 29.87 kg, *p* < 0.001). Greater height in obese children aligns with evidence suggesting the earlier pubertal development and accelerated growth associated with obesity, which are likely due to hormonal imbalances, such as increased insulin-like growth factor (IGF-1). HDL cholesterol was below 40 mg/dL in children with obesity and, as expected, significantly different from that of children with normal weight. The low levels of HDL cholesterol in children with MetS (32.67 mg/dL) and without MetS (41.85 mg/dL) in this study may indicate metabolic dysfunction, systemic inflammation, or physical inactivity. These findings are consistent with extensive literature linking obesity to cardiometabolic risk factors from childhood to adulthood. [27,28]. 

The presence of metabolic syndrome in children with obesity should not be minimized, even when there are different criteria for its diagnosis [29,30]. In our population, according to the IDF criteria, 55% of the children with obesity and 23.6% of the total children had MetS. Other studies from the center of Mexico have reported a prevalence of MetS between 2.4 and 45.9% in children 9–13 years of age [31]. In order to establish this magnitude, it is strongly suggested that globally standardized criteria be used for the definition of metabolic syndrome in children; currently, several criteria are considered in research and clinical practice, and higher proportions are reported by researchers using the de Ferranti criteria [31], thereby affecting the recorded prevalence.

Children with obesity presented leptin levels higher than those of children with normal BMIs. Serum leptin levels in children with MetS in the present study (32.42 ± 21.39 ng/mL) were twofold higher than those indicated in the results reported in Chinese children aged 6–18 years with three or more components of MetS (15.03 ng/mL in boys and 16.35 ng/mL in girls) [32]. The results could be explained according to those reported by Frühbeck et al. (2019), who stated that obesity may lead to an alteration of adipokine secretion, causing an ectopic accumulation of fat and lipotoxicity [33]. 

In our study, among school-aged children with obesity, whether they had MetS or not, there were no statistically significant differences in clinical and anthropometric characteristics, except for two MetS parameters, triglycerides and HDL cholesterol, which showed statistical significance according to the IDF criteria. It is worth noting that in our population, triglyceride levels exhibited considerable variability, ranging from a minimum of 150 mg/dL to a maximum of 524 mg/dL in the group with MetS, while HDL cholesterol levels ranged from 25 mg/dL (minimum) to 38 mg/dL (maximum). These findings suggest that even children without MetS already present clinical characteristics indicative of metabolic risk.

Although leptin levels are notably higher in children with MetS compared to those without MetS, this difference is not statistically significant. This may indicate that while leptin dysregulation is common in obesity, its role as a differentiator for MetS may be limited in the study group. Adiponectin levels tend to be lower in children with MetS, which aligns with its known anti-inflammatory and insulin-sensitizing properties. However, the lack of statistical significance suggests that further investigation with larger sample sizes may be needed.

No statistical differences were found in IL-6 and TNF-α between groups of children with obesity vs. those of normal weight or in children with vs. those without MetS. Similarly, others have also reported no significant difference in IL-6 levels in prepubertal children from India, at 3.56 vs. 3.76 pg/mL (*p* = 0.850) in the case (obesity) vs. control (normal weight) groups, respectively [34]. These results suggest that IL-6 and TNF-α are not directly related to levels of adiposity in children. However, others have recently reported differences in levels of the aforementioned adipokines among Argentinean children and adolescents (5–19 years), grouped into those with normal weight, overweight, and obesity, demonstrating higher levels of IL-6 in boys with obesity, at around 2.34 pg/mL (*p* = 0.028) [35]. Therefore, the need for more research into these particular adipokines is indicated. It is of interest to note that when comparing groups of obese children with vs. without MetS, the biomarker showing a statistical difference was serum gamma-glutamyl transferase (GGT) (*p* = 0.02). GGT levels are significantly higher in children without MetS. This unexpected finding warrants further exploration, as GGT is typically associated with oxidative stress and liver dysfunction, which are common in MetS. The data seen here could reflect unique pathophysiological dynamics in this pediatric population.

An assessment of the association between adipokines and GGT levels with the presence of MetS according to the study population characteristics resulted in a 27 times higher probability of presenting with metabolic syndrome when leptin levels were elevated (mean of 32.42 ng/mL), and low adiponectin levels (mean of 28.00 µg/mL) resulted in a 10 times higher probability of having the condition. The regression model demonstrates the importance of leptin and adiponectin as potential early indicators for identifying children at risk for MetS, offering opportunities for targeted interventions.

Other adipokines, such as TNF-α, IL-6, and GGT, were not significantly associated with the presence of MetS in this group of children. While studying the association with the components of MetS, it was observed that leptin, a well-known adipokine, had significant associations with parameters such as WC, SBP, glucose, triglycerides, and HDL cholesterol. These associations align with the existing literature that emphasizes leptin’s role in energy homeostasis and its involvement in the pathophysiology of obesity-related complications. The correlation between leptin and WC suggests that visceral fat accumulation may be a driving factor in the development of metabolic syndrome, emphasizing the importance of targeted interventions focused on reducing abdominal obesity in children. Adiponectin, which typically exhibits protective cardiovascular and metabolic effects, had negative associations with WC, SBP, and triglycerides and a positive association with HDL cholesterol. The inverse relationship between adiponectin levels and metabolic syndrome components, except for HDL cholesterol, supports its protective role, suggesting that lower adiponectin levels may predispose children to metabolic dysregulation. This finding reinforces the potential of adiponectin as a therapeutic target in managing pediatric obesity and its complications. Our study also presents GGT as a significant biomarker that is positively associated with WC and negatively associated with HDL cholesterol. GGT is involved in oxidative stress and inflammation, which identifies it as a relevant marker in metabolic syndrome, corroborating previous studies that associate elevated GGT levels with metabolic disturbances.

An association between adipokines and GGT was also determined, with observed significance among leptin, adiponectin, and GGT (*p* < 0.001), suggesting a complex interplay among these biomarkers in the context of metabolic syndrome. This interplay warrants further investigation to elucidate the mechanisms underlying these associations. The inflammatory marker TNF-α did not exhibit significant associations with the metabolic variables examined. This observation suggests that although inflammation contributes to complications associated with obesity, the direct impact of this cytokine could be more complex or vary depending on the stage of development in pediatric populations. Future studies should explore the temporal dynamics of inflammatory markers in conjunction with adipokines to better understand their contribution to metabolic syndrome. 

Lastly, the ROC curve and Youden’s index analyses provide valuable cut-off points for the studied biomarkers: 8.3665 ng/mL for leptin, 9.87 µg/mL for adiponectin, and 17.8 U/L for GGT. Few studies have suggested cut-off points for biomarkers; a recent study indicated a level of 19.5 U/L for GGT to indicate children and adolescents at risk of metabolic syndrome in Sri Lanka [36]. The studied and proposed thresholds can facilitate the early distinction of children at risk for metabolic syndrome, enabling timely interventions. By incorporating these biomarkers into routine clinical assessments, healthcare providers can enhance early detection and potentially improve outcomes for children struggling with obesity. Neglecting such screening could delay diagnosis, allowing metabolic abnormalities to progress and leading to increased healthcare burdens in adulthood. These markers are not just diagnostic tools but also pivotal for targeted public health strategies. 

## 5. Conclusions

This study highlights the potential of circulating adipokines and GGT as valuable biomarkers for identifying metabolic syndrome risk in obese children from Northeast Mexico. Our findings add to the growing evidence supporting the use of these markers in both clinical settings and research. The complexity of pediatric MetS justifies the need for more research into nontraditional biomarkers, as well as in larger and more diverse populations.

## Figures and Tables

**Figure 1 nutrients-16-04410-f001:**
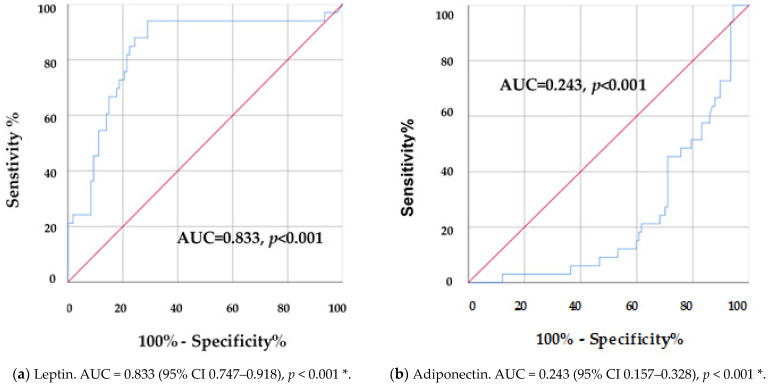
Area under the ROC curve (AUC) for (**a**) leptin, (**b**) adiponectin, (**c**) gamma-glutamyl transferase (GGT), and (**d**) interleukin-6 (IL-6). * Denotes significance when *p* < 0.05. ROC: receiving operating characteristics curve analysis. Area under the ROC curve (AUC) for (**e**) tumor necrosis factor alpha (TNF-α). * Denotes significance when *p* < 0.05. ROC: receiving operating characteristic curve analysis.

**Table 1 nutrients-16-04410-t001:** Clinical, anthropometric, and biochemical characteristics in children 6 to 12 years of age, according to body weight ^1^.

	Total (*n* = 140)	Children with Obesity (*n* = 60)	Children with Normal Weight (*n* = 80)	*p*
**Gender (n (%))**				
**Girls**	83 (59.3)	31 (51.7)	52 (65.0)	0.021 *
**Boys**	57 (40.7)	29 (48.3)	28 (35.0)	0.895
**Age (years)**	9.03 ± 1.87	9.55 ± 1.61	8.64 ± 1.96	0.003 *
**Height (cm)** ^a^	137.47 ± 14.24	144.78 ± 11.57	131.98 ± 13.63	<0.001 *
**Weight (kg)**	42.81 ± 18.67	60.07 ± 14.37	29.87 ± 7.96	<0.001 *
**BMI (kg/m** ** ^2^ ** **)**	21.71 ± 6.32	28.28 ± 3.89	16.78 ± 1.31	<0.001 *
**Z-score of BMI**	1.53 ± 1.66	3.29 ± 0.86	0.22 ± 0.47	<0.001 *
**WC (cm)**	71.91 ± 17.73	89.90 ± 11.09	58.42 ± 5.60	<0.001 *
**WHtR**	0.52 ± 0.09	0.62 ± 0.06	0.44 ± 0.03	<0.001 *
**SBP (mmHg)** ^a^	93.30 ± 12.07	99.05 ± 10.51	88.99 ± 11.40	<0.001 *
**DBP (mmHg)**	62.98 ± 7.52	64.20 ± 8.10	62.06 ± 6.96	0.001 *
**MAP (mmHg)** ^a^	73.09 ± 8.51	75.82 ± 8.38	71.04 ± 8.07	<0.001 *
**Glucose (mg/dL)** ^a^	83.66 ± 7.21	84.83 ± 7.28	82.79 ± 7.07	0.090
**Triglycerides (mg/dL)**	137.64 ± 66.81	175.92 ± 79.62	108.93 ± 34.25	<0.001 *
**HDL cholesterol (mg/dL)**	45.49 ± 12.89	36.80 ± 7.91	52.01 ± 12.06	<0.001 *

^1^ Numerical data are shown as media ± standard deviation; categorical data are shown as frequency and percentage (n (%)). BMI: body mass index; WC: waist circumference; WHtR: waist-to-height ratio; SBP systolic blood pressure; DBP: diastolic blood pressure; MAP: mean arterial pressure. A Student’s *t*-test was used for independent samples to compare groups of children with obesity vs. those of normal weight. ^a^ Equal variances were assumed according to Levene’s test. A chi-squared test was used to compare categorical data between groups. * Denotes significance when *p* < 0.05.

**Table 2 nutrients-16-04410-t002:** Clinical, anthropometric, and biochemical characteristics in obese children 6 to 12 years of age, according to the presence or absence of metabolic syndrome (MetS) ^1^.

	Children with MetS (*n* = 33)	Children without MetS (*n* = 27)	*p*
**Gender (n (%))**			
**Girls**	19 (57.6)	12 (44.4)	0.209
**Boys**	14 (42.4)	15 (55.6)	0.853
**Age (years)**	9.27 ± 1.57	9.89 ± 1.63	0.142
**Height (cm)** ^a^	143.12 ± 11.40	146.81 ± 11.67	0.220
**Weight (kg)**	58.01 ± 13.94	62.59 ± 14.76	0.220
**BMI (kg/m^2^)**	27.99 ± 4.04	28.64 ± 3.74	0.520
**Z-score of BMI**	3.30 ± 0.99	3.27 ± 0.68	0.890
**WC (cm)**	89.35 ± 11.55	90.58 ± 10.67	0.660
**WHtR**	0.62 ± 0.07	0.61 ± 0.06	0.647
**SBP (mmHg)** ^a^	98.48 ± 11.97	99.74 ± 8.56	0.640
**DBP (mmHg)**	65.00 ± 8.86	63.22 ± 7.09	0.400
**MAP (mmHg)** ^a^	76.16 ± 9.46	75.39 ± 6.99	0.728
**Glucose (mg/dL)** ^a^	85.30 ± 8.27	84.26 ± 5.95	0.580
**Triglycerides (mg/dL)**	216.21 ± 80.16	126.67 ± 43.54	<0.001 *
**HDL cholesterol (mg/dL)**	32.67 ± 3.93	41.85 ± 8.65	<0.001 *

^1^ Data are shown as media ± standard deviation. BMI: body mass index; WC: waist circumference; WHtR: waist-to-height ratio; SBP systolic blood pressure; DBP: diastolic blood pressure, MAP: mean arterial pressure. Metabolic syndrome was defined based on the criteria of the International Diabetes Federation (IDF) (2007) [7]. A Student’s *t*-test was used for independent samples to compare groups of children with MetS vs. without MetS. ^a^ Equal variances were assumed according to Levene’s test. A chi-squared test was used to compare categorical data between groups. * Denotes significance when *p* < 0.05.

**Table 3 nutrients-16-04410-t003:** Circulating serum levels of adipokines and GGT in children 6 to 12 years of age ^1^.

	Total (*n* = 140)	Children with Obesity (*n* = 60)	Children with Normal Weight (*n* = 80)	*p*
**Leptin (ng/mL)**	15.49 ± 18.04	30.61 ± 18.60	4.14 ± 3.19	<0.001 *
**Adiponectin (µg/mL)**	51.35 ± 39.85	29.54 ± 23.95	67.71 ± 41.63	<0.001 *
**GGT (U/L)**	17.94 ± 9.64	21.86 ± 13.49	15.00 ± 2.69	<0.001 *
**IL-6 (pg/mL)** ^a^	4.62 ± 4.90	5.11 ± 4.30	4.26 ± 5.30	0.310
**TNF-α (pg/mL)** ^a^	6.24 ± 4.14	6.64 ± 3.66	5.94 ± 4.47	0.320

^1^ Data are shown as media ± standard deviation. GGT: gamma-glutamyl transferase; IL-6: interleukin 6; TNF-α: tumor necrosis factor alpha. A Student’s *t*-test was used for independent samples to compare groups of children with obesity vs. those without obesity. ^a^ Equal variances were assumed according to Levene’s test. * Denotes significance when *p* < 0.05.

**Table 4 nutrients-16-04410-t004:** Circulating serum levels of adipokines and GGT in obese children 6 to 12 years of age, according to the presence or absence of metabolic syndrome (MetS) ^1^.

Adipokines and GGT	Children with MetS (*n* = 33)	Children without MetS (*n* = 27)	*p*
**Leptin (ng/mL)** ^a^	32.42 ± 21.39	28.40 ± 14.58	0.400
**Adiponectin (µg/mL)** ^a^	28.00 ± 19.98	31.42 ± 28.34	0.580
**GGT (U/L)**	18.34 ± 7.77	26.17 ± 17.43	0.020 *
**IL-6 (pg/mL)** ^a^	5.53 ± 4.90	4.59 ± 3.45	0.400
**TNF-α (pg/mL)** ^a^	6.71 ± 4.15	6.55 ± 3.04	0.870

^1^ Data are shown as media ± standard deviation. GGT: gamma-glutamyl transferase; IL-6: interleukin 6; TNF-α: tumor necrosis factor alpha. Metabolic syndrome was defined based on the criteria of the International Diabetes Federation (IDF) (2007) [7]. A Student’s *t*-test was used for independent samples to compare groups of children with MetS vs. those without MetS. ^a^ Equal variances were assumed according to Levene’s test. * Denotes significance when *p* < 0.05.

**Table 5 nutrients-16-04410-t005:** Simple linear regression model to analyze the association of metabolic syndrome with adipokines and GGT in obese children 6 to 12 years of age (*n* = 33) ^1^.

Dependent	Independent	R Squared	ANOVA	B	T	*p*
**Metabolic syndrome**	**Leptin (ng/mL)**	0.274	52.002	0.523	7.211	<0.001 *
	**Adiponectin (µg/mL)**	0.107	16.467	−0.327	−4.058	<0.001 *
	**GGT (U/L)**	0.001	0.072	0.023	0.268	0.789
	**IL-6 (pg/mL)**	0.011	1.484	0.103	1.218	0.225
	**TNF-α (pg/mL)**	0.004	0.547	0.063	0.740	0.461

^1^ GGT: gamma-glutamyl transferase; IL-6: interleukin 6; TNF-α: tumor necrosis factor alpha. Metabolic syndrome was defined based on the criteria of the International Diabetes Federation (IDF) (2007) [7]. A linear regression model was used. * Denotes significance when *p* < 0.05.

**Table 6 nutrients-16-04410-t006:** Linear regression model of components of metabolic syndrome (MetS), adipokines, and GGT in obese children 6 to 12 years of age (*n* = 33) ^1^.

	Leptin			Adiponectin			GGT		
MetS Components	R²	B	*p*	R²	B	*p*	R²	B	*p*
**WC**	0.452	0.684	<0.001 *	0.163	−0.908	<0.001 *	0.095	0.168	<0.001 *
**SBP**	0.110	0.496	<0.001 *	0.093	−1.005	<0.001 *	0.023	0.122	0.072
**DBP**	0.006	0.186	0.363	0.023	−0.802	0.074	< 0.001	−0.011	0.922
**Glucose**	0.035	0.469	0.026 *	0.003	−0.288	0.541	0.013	0.150	0.187
**Triglycerides**	0.130	0.097	<0.001 *	0.075	−0.164	0.001 *	0.003	−0.008	0.491
**HDL−cholesterol**	0.212	−0.644	<0.001 *	0.148	1.187	<0.001 *	0.054	−0.174	0.006 *

^1^ MetS: metabolic syndrome; WC: waist circumference; SBP systolic blood pressure; DBP: diastolic blood pressure, GGT: gamma-glutamyl transferase. Metabolic syndrome criteria of the International Diabetes Federation (IDF) (2007) [7]. A linear regression model was used. * Denotes significance when *p* < 0.05.

**Table 7 nutrients-16-04410-t007:** Linear regression model of components of metabolic syndrome (MetS) and adipokines (IL-6 and TNF-α) in obese children 6 to 12 years of age (*n* = 33) ^1^.

	IL-6			TNF-α		
MetS Components	R²	B	*p*	R²	B	*p*
**WC**	0.015	0.034	0.151	0.002	0.010	0.612
**SBP**	0.007	0.035	0.312	0.004	−0.022	0.444
**DBP**	<0.001	−0.005	0.921	0.006	−0.041	0.379
**Glucose**	0.004	−0.041	0.477	0.001	−0.020	0.688
**Triglycerides**	0.013	0.008	0.175	0.002	−0.003	0.630
**HDL cholesterol**	0.032	−0.067	0.036 *	0.018	−0.043	0.117

^1^ MetS: metabolic syndrome; WC: waist circumference; SBP systolic blood pressure; DBP: diastolic blood pressure; IL-6: interleukin 6; TNF-α: tumor necrosis factor alpha. Metabolic syndrome criteria of the International Diabetes Federation (IDF) (2007) [7]. A linear regression model was used. * Denotes significance when *p* < 0.05.

**Table 8 nutrients-16-04410-t008:** Association of adipokines and GGT ^1^.

	Leptin		
Markers	R²	B	*p*
**Adiponectin**	0.120	−0.157	<0.001 *
**TNF-α**	0.023	0.663	0.072
**IL-6**	0.018	0.499	0.110
**GGT**	0.113	0.629	<0.001 *
	**Adiponectin**		
	**R²**	**B**	** *p* **
**Leptin**	0.120	−0.766	<0.001 *
**TNF-α**	0.001	−0.229	0.780
**IL-6**	<0.001	0.175	0.801
**GGT**	0.088	−1.228	<0.001 *
	**TNF−α**		
	**R²**	**B**	** *p* **
**Leptin**	0.023	0.035	0.072
**Adiponectin**	0.001	−0.002	0.780
**IL-6**	0.092	0.256	<0.001 *
**GGT**	0.002	0.021	0.573
	**IL−6**		
	**R²**	**B**	** *p* **
**Leptin**	0.018	0.037	0.110
**Adiponectin**	<0.001	0.003	0.801
**TNF-α**	0.092	0.358	<0.001 *
**GGT**	<0.001	0.010	0.819
	**GGT**		
	**R²**	**B**	** *p* **
**Leptin**	0.113	0.180	<0.001 *
**Adiponectin**	0.088	−0.072	<0.001 *
**TNF-α**	0.002	0.112	0.573
**IL-6**	<0.001	0.038	0.573

^1^ IL-6: interleukin 6. TNF-α: tumor necrosis factor alpha. A linear regression model was used. * Denotes significance when *p* < 0.05.

**Table 9 nutrients-16-04410-t009:** Area under the ROC curve (AUC) for adipokines and GGT as predictors of metabolic syndrome ^1^.

Markers	AUC (95% CI)	*p*
**Leptin (ng/mL)**	0.833 (0.747 a 0.918)	<0.001 *
**Adiponectin (µg/mL)**	0.243 (0.157 a 0.328)	<0.001 *
**GGT (U/L)**	0.626 (0.508 a 0.744)	0.029 *
**IL-6 (pg/mL)**	0.647 (0.552 a 0.743)	0.011 *
**TNF-α (pg/mL)**	0.552 (0.437 a 0.667)	0.059

^1^ GGT: gamma-glutamyl transferase; IL-6: interleukin 6; TNF-α: tumor necrosis factor alpha. A receiver operating characteristic (ROC) curve analysis was used. * Denotes significance when *p* < 0.05.

**Table 10 nutrients-16-04410-t010:** Cut-off points obtained using Youden’s index ^1^.

Markers	Cut-Off Points
**Leptin (ng/mL)**	8.3665 ng/mL
**Adiponectin (µg/mL)**	9.87 µg/mL
**GGT (U/L)**	17.8 U/L
**IL-6 (pg/mL)**	2.77 pg/mL
**TNF-α (pg/mL)**	6.68 pg/mL

^1^ Youden’s index helps identify the threshold that maximizes the tes’t overall diagnostic effectiveness. GGT: gamma-glutamyl transferase; IL-6: interleukin 6; TNF-α: tumor necrosis factor alpha.

## Data Availability

Data are available upon request from the corresponding author. The data are not publicly available due to confidentiality and privacy concerns.

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
