# Peer review of "Adipokines and Gamma-Glutamyl Transferase as Biomarkers of Metabolic Syndrome Risk in Mexican School-Aged Children"

_nutrients, 2024, doi:10.3390/nu16244410_

Round 1

Reviewer 1 Report

Comments and Suggestions for Authors

The manuscript titled "Adipokines and Gamma-Glutamyl Transferase as Biomarkers of Metabolic Syndrome Risk in Mexican School-Aged Children" presents a relevant study addressing the increasing prevalence of metabolic syndrome in children, particularly in the context of Mexican populations.

The topic is highly relevant given the rising rates of obesity and metabolic syndrome among children, especially in Mexico. The focus on adipokines and gamma-glutamyl transferase (GGT) as potential biomarkers adds significant value to the field of pediatric health and nutrition.

Please include these modifications to improve the paper:1

1)     If you discuss implications for dietary strategies in managing metabolic syndrome or improving insulin sensitivity, citing this paper can provide additional evidence for your recommendations and you can cite this paper Rondanelli, M., et al & Donini, L. M. (2013). Improvement in insulin resistance and favourable changes in plasma inflammatory adipokines after weight loss associated with two months’ consumption of a combination of bioactive food ingredients in overweight subjects. Endocrine, 44, 391-401.

2)     In addition revise the introduction, clearly define terms like "adipokines" and "insulin resistance" early on to set a solid foundation for readers unfamiliar with the concepts.

3)     Provide more detailed information about the statistical methods used, including justification for choosing specific tests and how they relate to the research questions.

4)     Include a brief explanation of why certain statistical tests were selected (e.g., why a T-test was used over ANOVA) and clarify how assumptions for these tests were checked (normality, homogeneity of variance).

5)     Broaden the discussion section to include more context about how the findings relate to existing literature and their potential implications for clinical practice or public health.

Comments on the Quality of English Language

fine

Author Response

Reviewer 1

Comment 1: If you discuss implications for dietary strategies in managing metabolic syndrome or improving insulin sensitivity, citing this paper can provide additional evidence for your recommendations and you can cite this paper Rondanelli, M., et al & Donini, L. M. (2013). Improvement in insulin resistance and favourable changes in plasma inflammatory adipokines after weight loss associated with two months’ consumption of a combination of bioactive food ingredients in overweight subjects. Endocrine, 44, 391-401.

Response 1: We appreciate the suggestion to discuss dietary implications for managing metabolic syndrome and the potential benefits of bioactive food ingredients. However, we have decided not to include the recommended reference by Rondanelli, M., et al & Donini, L. M (2013), as our study does not focus on dietary interventions but rather on identifying biomarkers associated with metabolic syndrome risk in children. Additionally, the cited study was conducted in an adult population, which differs significantly in physiology and metabolism compared to the pediatric population addressed in our research. While dietary strategies are indeed important, they fall outside the scope of our current study. We have instead emphasized the relevance of our findings for early identification and prevention strategies in children.

Comment 2:    In addition revise the introduction, clearly define terms like "adipokines" and "insulin resistance" early on to set a solid foundation for readers unfamiliar with the concepts.

Response 2: It was added as suggested. From the revision of the introduction we added on the page 2, Paragraph 2, on lines 20 and 21 the next text: “and altered levels of  adipokines”. Also we added the definition of adipokines and insulin resistance to provide context for readers unfamiliar with these concepts on page 2,  Paragraph 3, lines 32-36.

“Adipose tissue secreted adipokines are a group of bioactive molecules that play critical roles in regulating different physiological and pathological processes, such as inflammation, metabolism, appetite, cardiovascular function, immunity, insulin sensitivity, among others [15, 16]. Insulin resistance is a condition whereby insulin-induce glucose uptake is impaired in the insulin-sensitivity tissue [17]”.

Comments 3: Provide more detailed information about the statistical methods used, including justification for choosing specific tests and how they relate to the research questions.

It was added as suggested, the statistical methodology section was expanded to include justifications for the chosen tests.  On page 4, Section 2.4, paragraph 2, between lines 9 to 21.

2.4 Statistical analysis

A normality test was run using the Kolmogorov-Smirnov test, obtaining normal distributions.  Descriptive data was shown as media and standard deviation (SD) and categorical data was presented as frequency and percentage (n(%)). A Student’s t-test for independent samples was run to compare clinical, biochemical and anthropometric parameters between two groups of children (with vs without obesity and those with vs without metabolic syndrome) [22], Levene´s test was considered for equal variances assumption.  Chi-squared test was used to compare categorical data from groups.

A linear regression model was used to determine the association between adipokines and GGT and the presence of metabolic syndrome, thus possibly state the former as predictive factor to the disease [23].  Lastly, a receiver operating characteristic (ROC) curve analysis was presented, the area under the ROC curve (AUC) was also represented and the Youden’s Index were estimated to establish reference values (cut-off points) of the markers under study that may help predict the development of metabolic syndrome in children between 6 to 12 years [24].

Statistical analyses were done using SPSS® Statistics version 25 (IBM® SPSS®), considering a p value < 0.05 as significant

Comments 4:    Include a brief explanation of why certain statistical tests were selected (e.g., why a T-test was used over ANOVA) and clarify how assumptions for these tests were checked (normality, homogeneity of variance).

Response 4: The T-test was used over ANOVA because the comparison was from two groups (independent samples); normality was previously obtained using the Kolmogorov-Smirnov test, the Levene´s test was considered for equal variances assumption. For the case of categorical variables, such as gender, Chi-squared test was used to compare the two groups.

It was added as suggested, the justifications for the chosen tests was reflected On page 4, Section 2.4, paragraph 2, between lines 12 to 16. Additionally, to improve the explanation for this comment, we added the statistical test as a footnote in tables 1, 2, 3 and 4.

2.4 Statistical analysis

A normality test was run using the Kolmogorov-Smirnov test, obtaining normal distributions.  Descriptive data was shown as media and standard deviation (SD) and categorical data was presented as frequency and percentage (n(%)). A Student’s t-test for independent samples was run to compare clinical, biochemical and anthropometric parameters between two groups of children (with vs without obesity and those with vs without metabolic syndrome) [22], Levene´s test was considered for equal variances assumption.  Chi-squared test was used to compare categorical data from groups.

Comments 5: Broaden the discussion section to include more context about how the findings relate to existing literature and their potential implications for clinical practice or public health.

Response 5: We have enriched the discussion section to provide a broader context, linking findings to relevant literature and highlighting their implications for clinical practice and public health. We align the text of the Discussion section on

page 13.

Paragraph 1, lines 8-12;

Paragraph 2, lines 13-21;

Paragraph 3, lines 24-23;

Paragraph 4, lines 33-41

page 14.

Paragraph 6,  on lines 8-16;

Paragraph 7, lines 17-23,

Paragraph 8, lines 36-39;

paragraph 9, on lines 44-46,

Page 15, paragraph 10, on lines 7-11

Paragraph 12, lines 34-36

Alarming data in obesity and metabolic syndrome and growing incidence at very young individuals, such as school–aged children, is considered nowadays as a health problem of high relevance for professionals and researchers in multidisciplinary settings. The identification of nontraditional biomarkers in clinical practice is pertinent for the management and prevention of complications in children with these conditions. Therefore, this concern has motivated to examine the association between circulating leptin, adiponectin and GGT with metabolic syndrome in school aged children from the Northeast Mexico.

When comparing clinical, anthropometric and biochemical characteristics of children with obesity against those with normal BMI (with normal weight) all the parameters were significantly different, except for glucose levels at 84.83 vs 82.79 mg/ml (p=0.09) and Diastolic Blood Pressure (64.20 vs 62.06 mmHg (p=0.10), respectively. This discrepancy may reflect the younger age group (6–12 years), where insulin resistance and hyperglycemia might not yet be prominent due to shorter obesity duration. Most findings confirm established evidence; however, the lack of glucose elevation may highlight the importance of early intervention before metabolic dysfunction becomes more pronounced in older children or adolescents [25,26].

In our population, height was higher in children in the group of obesity (144.78 cm) vs that in normal weight children (131.98 cm); and children with obesity presented twice the body weight of those with normal weight (60.07 kg vs 29.87 kg, p<0.001). Higher height in obese children aligns with evidence suggesting earlier pubertal development and accelerated growth associated with obesity, likely due to hormonal imbalances such as increased insulin-like growth factor (IGF-1). HDL-cholesterol was below 40 mg/dL in children with obesity and, as expected, significantly different from children with normal weight. The low level of HDL-cholesterol in children with MetS (32.67 mg/dL) and without MetS (41.85 mg/dL) of this study may indicate metabolic dysfunction, systemic inflammation, or physical inactivity. These findings are consistent with extensive literature linking obesity to cardiometabolic risk factors from childhood to adulthood. [27, 28].  

 The presence of metabolic syndrome in children with obesity should not be minimized, even when there are different criteria for its diagnosis [29, 30]. In our population, with the IDF criteria, 55% of the children with obesity and 23.6% from the total children had MetS. Other studies from the center of Mexico have reported a prevalence of MetS between 2.4 and 45.9% in children 9–13 years [31].  In order to establish this magnitude it is strongly suggested a global standardized criteria for the definition of metabolic syndrome in children, as it affects prevalence as several criteria are considered in research and clinical practice, reporting higher proportions while using the De Ferranti criteria [31].

Children with obesity presented leptin levels above those children with a normal BMI. Serum leptin levels in children with MetS found in the present study (32.42 ± 21.39 ng/mL) were two-fold higher than results reported in Chinese children aged 6–18 years with 3 or more components of MetS (15.03 ng/mL in boys and 16.35 ng/mL in girls) [32]. Results could be explained according to that reported by Frühbeck et al (2019), stating that obesity may lead to an alteration of adipokine secretion, causing an ectopic accumulation of fat and lipotoxicity [33]. 

Although leptin levels are notably higher in children with MetS compared to those without MetS, this difference is not statistically significant. This may indicate that while leptin dysregulation is common in obesity, its role as a differentiator for MetS may be limited in the study group. Adiponectin levels tend to be lower in children with MetS, which aligns with its known anti-inflammatory and insulin-sensitizing properties. However, the lack of statistical significance suggests that further investigation with larger sample sizes may be needed.

No statistical differences were found in IL-6 and TNF-α between groups of children with obesity vs normal weight, and in those levels in children with vs without MetS.  Similarly, others have also reported no significant difference in IL-6 levels in prepubertal children from India at 3.56 vs 3.76 pg/mL (p=0.850), in case (obesity) vs control (normal weight) groups, respectively [34]. These results have led to suggest that IL-6 and TNF-α are not directly related to levels of adiposity in children. However, others have recently reported differences in levels in the mentioned adipokines among Argentinean children and adolescents (5–19 years) grouped into normal weight, overweight and with obesity, demonstrating higher levels in boys with obesity for IL-6 at around 2.34 pg/mL (p=0.028) [35]. Therefore, more research is suggested in these particular adipokines.  It is of interest to note that when comparing groups of obese children with vs without MetS, the biomarker showing statistical difference was serum gamma-glutamyl transferase (GGT) (p=0.02). GGT levels are significantly higher in children without MetS. This unexpected finding warrants further exploration, as GGT is typically associated with oxidative stress and liver dysfunction, which are common in MetS. The data seen here could reflect unique pathophysiological dynamics in this pediatric population.

An assessment of the association between adipokines and GGT levels with the presence of MetS, according to the study population characteristics, resulted in 27 times more possibility for presenting metabolic syndrome when leptin levels were elevated (mean of 32.42 ng/mL) and low adiponectin levels (mean of 28.00 µg/mL) resulted in 10 times more possibility of having the condition. The regression model highlights the importance of leptin and adiponectin as potential early indicators for identifying children at risk for MetS, offering opportunities for targeted interventions.

Other adipokines such as TNF-α, IL-6 and GGT did not show significant association with the presence of MetS in this group of children.  While studying the association with the components of MetS, it was observed that leptin, a well-known adipokine, showed a significant association with parameters such as WC, SBP, glucose, triglycerides and HDL-cholesterol.  These associations align with existing literature that highlights leptin's role in energy homeostasis and its involvement in the pathophysiology of obesity-related complications. The correlation between leptin and WC suggests that visceral fat accumulation may be a driving factor in the development of metabolic syndrome, emphasizing the importance of targeted interventions focused on reducing abdominal obesity in children.  Adiponectin, which typically exhibits protective cardiovascular and metabolic effects, showed negative associations with WC, SBP, triglycerides and a positive with HDL-cholesterol. The inverse relationship between adiponectin levels and metabolic syndrome components, except for HDL-cholesterol, supports its protective role, suggesting that lower adiponectin levels may predispose children to metabolic dysregulation. This finding reinforces the potential of adiponectin as a therapeutic target in managing pediatric obesity and its complications. Our study also highlights GGT as a significant biomarker positively associated with WC and negatively with HDL-cholesterol. GGT is involved in oxidative stress and inflammation, this reflects it as a relevant marker in metabolic syndrome, corroborating previous studies that associate elevated GGT levels with metabolic disturbances. 

Association among adipokines and GGT were also determined, observing significance among leptin, adiponectin and GGT (p<0.001) suggest a complex interplay between these biomarkers in the context of metabolic syndrome. This interplay warrants further investigation to elucidate the mechanisms underlying these associations.  The inflammatory marker TNF-α did not exhibit significant associations with the metabolic variables examined. This observation suggests that although inflammation contributes to complications associated with obesity, the direct impact of this cytokine could be more complex or vary depending on the stage of development in pediatric populations. Future studies should explore the temporal dynamics of inflammatory markers in conjunction with adipokines to better understand their contribution to metabolic syndrome.

Lastly, the ROC curve and Youden’s index analyses provide valuable cut-off points for the identified biomarkers: leptin at 8.3665 ng/mL, adiponectin at 9.87 µg/mL, and GGT at 17.8 U/L. Few studies have suggested cut-off points for biomarkers, a recent study stated a level of GGT at 19.5 U/L to identify children and adolescents at risk of metabolic syndrome in Sri Lanka [36]. The studied and proposed thresholds can facilitate early identification of children at risk for metabolic syndrome, enabling timely interventions. By incorporating these biomarkers into routine clinical assessments, healthcare providers can enhance early detection and potentially improve outcomes for children struggling with obesity. Neglecting such screening could delay diagnosis, allowing metabolic abnormalities to progress and leading to increased healthcare burdens in adulthood. These markers are not just diagnostic tools but also pivotal for targeted public health strategies.

Reviewer 2 Report

Comments and Suggestions for Authors

Several things need to be added or changed in the paper:

In the introduction

There should be a description of the relationships between the factors studied, with a figure to show them.

The aim needs to be changed. It is not possible to write about the population of Mexico as a whole if you are studying people from one region.

In the methods 

Give criteria for excluding children from other regions.

In the results 

Please give the WHR and MAP.

In Table 1, please include data on age and sex.

Table 5 has no exploratory value, such data are known from the literature.

Table 6 and others, you cannot write a statistical significance result of 0.000. Please write what exactly the result of the analysis was. Or check that such a result is not an error due to poor preparation of the data for statistical analysis.

The MetS+ and MetS- groups are not statistically different from each other on the key parameters that define MetS. What was the criterion used to create a MetS+ group?

Please explain what the 'AUC' in the graphs means.

In discussion

Some of the discussion has been repeated from the introduction, including the quotation. Please revise this.

No data or discussion on how exercise changes inflammatory parameters. Was this parameter considered? Such low HDL in both groups indicates something.

The conclusions are not consistent with the stated aims.

The conclusions are completely unjustified and exaggerated. There are 60 people with metabolic syndrome, far too few to draw such far-fetched conclusions and call them a discovery.

What are the practical implications of the study? You don't need a study like this to know that obesity is a clinical problem. How can this research help?

Author Response

Reviewer 2

Comments 1:  In the introduction

There should be a description of the relationships between the factors studied, with a figure to show them.

Response 1: We appreciate this suggestion regarding the inclusion of a description of the relationships between the factors studied. We would like to clarify that these relationships are already explained in the text of the introduction, providing a detailed overview of how the studied biomarkers interact with metabolic syndrome. Additionally, we plan to adapt and visually represent these relationships in the proposed visual abstract to enhance clarity and accessibility for readers.

Comments 2:  The aim needs to be changed. It is not possible to write about the population of Mexico as a whole if you are studying people from one region.

Response 2: The objective was changed as suggested to specify the regional focus, On the Page 3 paragraph 1, line 1-3

“This study aims to examine the association of circulating adipokines and GGT with metabolic syndrome risk in school-aged children (6 to 12 years) from Northeast Mexico.” 

Comments 3.- In the methodsGive criteria for excluding children from other regions.

Response 3: Clarifications on exclusion criteria were added to the Methods section, specifying that children outside Nuevo León were not included because they did not attend the obesity program. Page 3, paragraph 3, line 20-22.

“School-aged children outside the state of Nuevo León do not attend the Obesity program, so they were not considered in the study”. 

Comments 4:  In the results Please give the WHR and MAP   

Response 4: Waist-to-Height Ratio (WHtR) and Mean Arterial Pressure (MAP) data were incorporated into Tables 1 and 2

Comments 5:  In Table 1, please include data on age and sex.

Response 5:  We added data on age and gender into Table 1 and we consider to add into table 2 this data  in order to improve the interpretation of the results in the groups of MetS+ and MetS-

Comments 6- Table 5 has no exploratory value, such data are known from the literature.

Response 6:  While it is true that some associations between metabolic syndrome (MetS) and biomarkers such as leptin and adiponectin have been reported in the literature, there is a lack of evidence for GGT, the added value of our findings lies in their focus on a population-specific context.

Moreover, the linear regression model presented in Table 5 points out statistically significant associations, confirming that higher levels of leptin and lower levels of adiponectin are strongly linked to an increased likelihood of MetS in this group (p < 0.001 for both). These findings not only reinforce existing knowledge but also validate its applicability in a population where such studies are scarce.

Additionally, your observation prompted us to carefully review the data in Table 5. Upon revisiting the table, we identified an error in the placement of negative signs, which has now been corrected in this revised version. We thank you for bringing this to our attention, as it allowed us to ensure the accuracy of our results.

Table 5: Simple linear regression model to analyze association of metabolic syndrome with adipokines and GGT in obese children 6 to 12 years (n=33).1

Dependent

Independent

R squared

ANOVA

 B

T

 p

Metabolic syndrome

Leptin (ng/mL)

Adiponectin (µg/mL)

GGT (U/L)

IL-6 (pg/mL)

TNF-α (pg/mL)

0.274

0.107

0.001

0.011

0.004

52.002

16.467

0.072

1.484

0.547

 0.523

-0.327

 0.023

  0.103

  0.063

 7.211

-4.058

0.268

1.218

0.740

<0.001*

<0.001*

 0.789

 0.225

 0.461

1 GGT: Gamma-Glutamyl Transferase; IL-6: Interleukin 6; TNF-α: Tumor necrosis factor alpha. Metabolic syndrome was defined based on the criteria of the International Diabetes Federation (IDF) (2007) [7]. Linear regression model was used. * Denotes significance when p<0.05.

The importance of evaluating the independent association of biomarkers with MetS in this pediatric group was explained in the Discussion section, paragraph9, Page 14, lines 44-46.

The regression model highlights the importance of leptin and adiponectin as potential early indicators for identifying children at risk for MetS, offering opportunities for targeted interventions.

Comments 7: Table 6 and others, you cannot write a statistical significance result of 0.000. Please write what exactly the result of the analysis was. Or check that such a result is not an error due to poor preparation of the data for statistical analysis.

Response 7: The inappropriate representation of p-values (e.g., 0.000) only on table 6 was corrected, ensuring precise reporting on Table 6, page 8, lines 18-24.

Table 6: Linear regression model of components of metabolic syndrome (MetS), adipokines and GGT in obese children 6 to 12 years (n=33).1

Leptin

Adiponectin

GGT

MetS

components

 R²

 B

 p

 B

 P

 R²

  B

  p

WC 

SBP

DBP

Glucose

Triglycerides

HDL-cholesterol

0.452

0.110

0.006

0.035

0.130

0.212

0.684

0.496

0.186

0.469

0.097

-0.644

<0.001*

<0.001*

0.363

0.026*

<0.001*

<0.001*

0.163

0.093

0.023

0.003

0.075

0.148

-0.908

-1.005

-0.802

-0.288

-0.164

1.187

<0.001*

<0.001*

0.074

0.541

0.001*

<0.001*

0.095

0.023

0.000

0.013

0.003

0.054

0.168

0.122

-0.011

0.150

-0.008

-0.174

<0.001*

 0.072

 0.922

 0.187

 0.491

 0.006*

1 MetS: metabolic syndrome; WC: waist circumference; SBP systolic blood pressure; DBP: diastolic blood pressure, GGT: Gamma-Glutamyl Transferase. Metabolic syndrome criteria of the International Diabetes Federation (IDF) (2007) [7]. Linear regression model was used. * Denotes significance when p<0.05.

Comments 8: The MetS+ and MetS- groups are not statistically different from each other on the key parameters that define MetS. What was the criterion used to create a MetS+ group?

Response 8: The criteria used to define Metabolic Syndrome (MetS) in this study were based on the International Diabetes Federation (IDF) guidelines for children and adolescents aged 10 to 16 years (IDF, 2007). According to these criteria, MetS is diagnosed if abdominal obesity (waist circumference ≥90th percentile, equivalent to ≥90 cm for boys and ≥80 cm for girls) is present, along with two or more of the following components:

Triglycerides ≥150 mg/dL

HDL-Cholesterol <40 mg/dL

Systolic Blood Pressure ≥130 mmHg or Diastolic Blood Pressure ≥85 mmHg

Fasting Blood Glucose ≥100 mg/dL

Regarding the results shown in Table 2, although most key components of MetS did not show statistically significant differences between the MetS+ and MetS- groups, significant differences were observed in Triglycerides and HDL-Cholesterol levels. Specifically, in the MetS+ group, Triglyceride levels ranged from 150 mg/dL (minimum) to 524 mg/dL (maximum), while HDL-Cholesterol levels ranged from 25 mg/dL (minimum) to 38 mg/dL (maximum). These findings are consistent with the IDF criteria and support the appropriateness of their application in defining the MetS+ group, then we included as footnote on the Table 2

Additionally, the criteria used to define MetS (IDF 2007) have been included as footnotes in the relevant tables for clarity:

  • Table 2: Page 6, Footnote Lines 6-7
  • Table 4: Page 7, Footnote Lines 23-24
  • Table 5: Page 8, Footnote Lines 5-6
  • Table 6: Page 8, Footnote Line 22
  • Table 7: Page 9, Footnote Line 7

Comments 9: Please explain what the 'AUC' in the graphs means.

Response 9: An explanation of "Area Under the Curve (AUC)" was included in the graphs' captions and relevant sections of the text for clarity such as:

Section 2.4  (Material and Methods) statistical analysis, Page 4 , paragraph 3,  line18 - 19

the area under the ROC curve (AUC) was also represented and the Youden’s Index were estimated to establish reference values (cut-off points)

Section 3.5 (Results) Cut-off points, Page 9, lines 21-23.

A receiver operating characteristic (ROC) curve analysis and area under the ROC curve (AUC) were represented for each adipokine and GGT (Figure 1, Table 9) to determine cut-off points for the prediction of metabolic syndrome in children. Areas under the ROC curve (AUC) were for leptin 0.833 (CI 95%, 0.747–0.918, p<0.001), adiponectin 0.243 (CI 95%, 0.157–0.328, p<0.001), GGT 0.626 (CI 95%, 0.508–0.744, p=0.029), IL-6 0.647 (CI 95%, 0.552–0.743, p=0.011), TNF-α 0.552 (CI 95%, 0.437–0.667, p=0.059).

 Figure 1, Page 11, line 2 and in Figure 1 (cont.) Page 12 line 4;

Figure 1. Area under the ROC curve (AUC) for (a) Leptin, (b) Adiponectin, (c) Gamma Glutamyl Transferase (GGT) and (d) Interleukin-6 (IL-6). * Denotes significance when p <0.05. ROC: receiving operating characteristics curve analysis.

Table 9, page12, line 8.

Table 9. Area under the ROC curve (AUC) for adipokines and GGT as predictors of metabolic syndrome.1

Comments 10: In discussion

Some of the discussion has been repeated from the introduction, including the quotation. Please revise this.

Response 10: Repetition from the Introduction in the Discussion section was removed, and this section was revised and corrected to focus on interpretation and implications of the findings.

Text removed

Introduction:

It has also been reported that high GGT levels may be a risk factor for the development of Metabolic Dysfunction Associated Steatotic Liver Diseases [11].

Discussion:

Paragraph 1

Therefore, this concern has motivated the development of the present study in children of 6–12 years of the state of Nuevo León in Mexico.  In the present study, 43% of children presented obesity in the northeast of Mexico, a percentage above that reported for combined overweight and obesity in Mexican children (5–11 years), at 33.2% by Pérez and Cruz in 2018 [1] and 36.5% by the ENSANUT in 2020-2024 [2], and by the ENSANUT in 2022 for Nuevo León state, at 34.2% [3].  A recent study has reported a presence of obesity in 37.5% of children and adolescents (8–15 years) in the east coast of Mexico [22].

Paragraph 2

A study in Taiwanese children (7–18 years) reported similar results between adipose vs normal weight groups, in terms of lack of statistical difference on parameters such as height (p=0.11), glucose levels (p=0.234) and DBP (p=0.063) [23].

Paragraph 3

Results from nutritional assessment of the study population demonstrate a health problem of high relevance which involves children requiring immediate diagnosis and treatment.

Paragraph 4

A study in Nuevo Leon state demonstrated a prevalence of MetS in children (6–18 years) with combined overweight and obesity at 58.81%; although, the de-Ferranti criteria was employed [6]. 

Paragraph 5

In addition, obesity causes adipose tissue to be dysfunctional, which is highly related with development of MetS, altering the secretion of adipokines that could be beneficial or detrimental for human health, of which the most important are leptin and adiponectin [16], as suggested by findings herein.

Paragraph 6

however, higher levels were found in children without MetS.  Park et al (2017) reported that levels of GGT between 15 and 19 U/L are related with metabolic disorders such as MetS and explained their association due to a chronic low-grade inflammatory state and oxidative stress [10]; in the present study, GGT levels in children with obesity were higher than those mentioned above, at levels of 18.34 and 26.17 U/L, in groups with and without MetS, respectively. Elevated levels of serum GGT suggests oxidative stress [14] and a chronically elevation in visceral adipose fat [31], such as hepatic fat [10], which may lead to the development of hyperglycemia, hypertriglyceridemia, hypertension [10,14] and a decrease in HDL-cholesterol [10], all being components of MetS

Paragraphs changed

Introduction

Adipose tissue secreted adipokines…… (we added the definition of adipokin and insulin resistance according to the suggestion of reviewer 1)……. 

The presence of metabolic syndrome in children with obesity should not be minimized, even when there are different criteria for its diagnosis [25, 26]. In our population, with the IDF criteria, 55% of the children with obesity and 23.6% from the total children had MetS. Other studies from the center of Mexico have reported a prevalence of MetS between 2.4 and 45.9% in children 9–13 years [24].  In order to establish this magnitude it is strongly suggested a global standardized criteria for the definition of metabolic syndrome in children, as it affects prevalence as several criteria are considered in research and clinical practice, reporting higher proportions while using the De Ferranti criteria [26].

Paragraph Added

Discussion

In our study, among school-aged children with obesity, whether they had MetS or not, there were no statistically significant differences in clinical and anthropometric characteristics, except for two MetS parameters: triglycerides and HDL cholesterol, which did show statistical significance according to the IDF criteria. It is worth noting that in our population, triglyceride levels exhibited considerable variability, ranging from a minimum of 150 mg/dL to a maximum of 524 mg/dL in the group  with MetS, while HDL cholesterol levels ranged from 25 mg/dL (minimum) to 38 mg/dL (maximum). These findings suggest that even children without MetS already present clinical characteristics indicative of metabolic risk.

Although leptin levels are notably higher in children with MetS compared to those without MetS, this difference is not statistically significant. This may indicate that while leptin dysregulation is common in obesity, its role as a differentiator for MetS may be limited in the study group. Adiponectin levels tend to be lower in children with MetS, which aligns with its known anti-inflammatory and insulin-sensitizing properties. However, the lack of statistical significance suggests that further investigation with larger sample sizes may be needed.

Paragraph

GGT levels are significantly higher in children without MetS. This unexpected finding warrants further exploration, as GGT is typically associated with oxidative stress and liver dysfunction, which are common in MetS. The data seen here could reflect unique pathophysiological dynamics in this pediatric population.

Neglecting such screening could delay diagnosis, allowing metabolic abnormalities to progress and leading to increased healthcare burdens in adulthood. These markers are not just diagnostic tools but also pivotal for targeted public health strategies.

Comments 11: No data or discussion on how exercise changes inflammatory parameters. Was this parameter considered? Such low HDL in both groups indicates something.

Rersponse 11:

While exercise data were not collected, we acknowledge its role in modulating inflammation and discuss the implications of low HDL levels in both groups.

Section discussion Page 13, paragraph 3, lines 27-32

HDL-cholesterol was below 40 mg/dL in children with obesity and, as expected, significantly different from children with normal weight. The low level of HDL-cholesterol in children with MetS (32.67 mg/dL) and without MetS (41.85 mg/dL) of this study may indicate metabolic dysfunction, systemic inflammation, or physical inactivity. These findings are consistent with extensive literature linking obesity to cardiometabolic risk factors from childhood to adulthood.

Comments 12: The conclusions are not consistent with the stated aims.

The conclusions are completely unjustified and exaggerated. There are 60 people with metabolic syndrome, far too few to draw such far-fetched conclusions and call them a discovery.

What are the practical implications of the study? You don't need a study like this to know that obesity is a clinical problem. How can this research help?

Response 12: The conclusions were revised to align with the study's scope and aims. We deleted a paragraph to avoid exaggeration and we highlighted the study's value in identifying early biomarkers and their potential role in clinical practice.

Section of Conclusions, Page 15, lines 38-43

The study highlights the potential of circulating adipokines and GGT as valuable biomarkers for identifying metabolic syndrome risk in obese children from the Northeast of Mexico. Our findings add to the growing evidence supporting the use of these markers in both clinical settings and research. The complexity of pediatric MetS justify the need for more research in the nontraditional biomarkers as well in larger and more diverse populations.

Reviewer 3 Report

Comments and Suggestions for Authors

How to explain both adipokines adiponectin and leptin secreted from fat tissue, high BMI subjects have more fat tissue, but the adiponectin in high BMI children is much lower but leptin’s much higher? What are the mechanisms behind these observations in previous studies?

In obese children, almost all the measurements in without MetS are higher, WC, weight, SBP and so on. It’s worth to discuss in detail.

Author Response

Reviewer 3

Comments 1: How to explain both adipokines adiponectin and leptin secreted from fat tissue, high BMI subjects have more fat tissue, but the adiponectin in high BMI children is much lower but leptin’s much higher? What are the mechanisms behind these observations in previous studies?

Response 1: The contrasting roles of adiponectin and leptin were further discussed, referencing previous studies to explain why leptin levels are elevated while adiponectin levels are reduced in obese children

Section Discussion  Page 14, lines 17-23.

“Although leptin levels are notably higher in children with MetS compared to those without MetS, this difference is not statistically significant. This may indicate that while leptin dysregulation is common in obesity, its role as a differentiator for MetS may be limited in the study group. Adiponectin levels tend to be lower in children with MetS, which aligns with its known anti-inflammatory and insulin-sensitizing properties. However, the lack of statistical significance suggests that further investigation with larger sample sizes may be needed”

Section Discussion  Page 14, lines 36-39

GGT levels are significantly higher in children without MetS. This unexpected finding warrants further exploration, as GGT is typically associated with oxidative stress and liver dysfunction, which are common in MetS. The data seen here could reflect unique pathophysiological dynamics in this pediatric population.

Comments 2.- In obese children, almost all the measurements in without MetS are higher, WC, weight, SBP and so on. It’s worth to discuss in detail.

Response 2: The observation that most measurements are higher in children with obesity but without MetS was elaborated in the Discussion section, emphasizing the importance of early metabolic changes

Section Discussion, Page 13,  Paragraph 3, lines 24-27

Higher height in obese children aligns with evidence suggesting earlier pubertal development and accelerated growth associated with obesity, likely due to hormonal imbalances such as increased insulin-like growth factor (IGF-1).